

# Identification of two functional xyloglucan galactosyltransferase homologs *BrMUR3* and *BoMUR3* in brassicaceous vegetables

Meng Wang[1,*], Zongchang Xu[2,*], Shuaiqiang Guo[1], Gongke Zhou[1], Malcolm ONeill[3] and Yingzhen Kong[1]

[1] College of Agronomy, Qingdao Agricultural University, Qingdao, China
[2] Marine Agriculture Research Center, Tobacco Research Institute, Chinese Academy of Agricultural Sciences, Qingdao, China
[3] Complex Carbohydrate Research Center, University of Georgia, Athens, GA, USA
[*] These authors contributed equally to this work.

## ABSTRACT

Xyloglucan (XyG) is the predominant hemicellulose in the primary cell walls of most dicotyledonous plants. Current models of these walls predict that XyG interacts with cellulose microfibrils to provide the wall with the rigidity and strength necessary to maintain cell integrity. Remodeling of this network is required to allow cell elongation and plant growth. In this study, homologs of *Arabidopsis thaliana MURUS3* (*MUR3*), which encodes a XyG-specific galactosyltransferase, were obtained from *Brassica rapa* (*BrMUR3*) to *Brassica oleracea* (*BoMUR3*). Genetic complementation showed that *BrMUR3* and *BoMUR3* rescue the phenotypic defects of the *mur3-3* mutant. Xyloglucan subunit composition analysis provided evidence that *BrMUR3* and *BoMUR3* encode a galactosyltransferase, which transfers a galactose residue onto XyG chains. The detection of XXFG and XLFG XyG subunits (restoration of fucosylated side chains) in *mur3-3* mutants overexpressing *BrMUR3* or *BoMUR3* show that MUR3 from *Brassica* to *Arabidopsis* are comparable as they add Gal to the third xylosyl residue of the XXXG subunit. Our results provide additional information for functional dissection and evolutionary analysis of *MUR3* genes derived from brassicaceous species.

Corresponding author
Yingzhen Kong,
kongyzh@qau.edu.cn

## INTRODUCTION

Xyloglucan (XyG) is present in the primary cell walls of land plants including gymnosperms, angiosperms, monilophytes, lycophytes, hornworts, mosses and liverworts (*Hoffman et al., 2005*; *Pena et al., 2008*; *Popper, 2008*; *Popper & Fry, 2003*, *2004*). It has a 1,4-linked β-glucan backbone that is, substituted at *O*-6 to varying extents with α-Xyl residues (*Pauly et al., 2013*). XyGs is also a storage polysaccharides in the seeds of several plant species (*Buckeridge, 2010*). In the primary wall XyGs is believed to associate with cellulose to prevent aggregation of cellulose microfibrils and thereby enable cellulose to interact with other cell wall components (*Pauly et al., 2013*).
Xyloglucan are synthesized in the Golgi apparatus (*Moore & Staehelin, 1988*). The glycosyltransferases including AtCSLC4, AtCSLC5 and AtCSLC6, which are encode by the cellulose synthase-like C (CSLC) gene family, are predicted to be involved in glucan backbone synthesis (*Cocuron et al., 2007*; *Zabotina, 2012*). *Xyloglucan Xylosyltransferase* (*XXT*) genes including *AtXXT1*, *AtXXT2* and *AtXXT5* catalyze the xylosylation of the glucan backbone (*Cavalier & Keegstra, 2006*; *Culbertson et al., 2016*; *Zabotina et al., 2012*). *Xyloglucan L-Side Chain Galactosyltransferase2* (*XLT2*) and *Murus3* (*MUR3*) are two distinct galactosyltransferases that transfer a galactosyl moiety to a different xylosyl residue (*Jensen et al., 2012*; *Madson et al., 2003*). *Fucosyltransferase1* (*FUT1*) encodes the fucosyltransferase that catalyzes the fucosylation of the Gal residue added by *MUR3* (*Perrin et al., 1999*).

Xyloglucan subunit structures are described by a single-letter nomenclature (*Fry et al., 1993*; *Tuomivaara et al., 2015*). For example, an unsubstituted backbone Glc residue is has the letter G, whereas side chains with xylosyl residues attached to Glc are indicated by the letter X. The addition of Gal, Ara, or Xyl to this Xyl residue is denoted by L, S (or D) and U, respectively. The addition of a Fuc residue to the Gal or Ara is shown as F and E, respectively. For many seed-bearing plants, their XyG has a XXXG-type structure in which three consecutive backbone Glc residues are substituted. In the model plant *A. thaliana*, the relative abundance of XXXG, XXFG, XLFG and XXLG is 1.0:1.0:1.7:0.3 (*Kong et al., 2015*; *Von Schantz et al., 2009*).

Mutations affecting XyG glycosyltransferase genes in *Arabidopsis* change the structure or content of XyG. For example, the XLFG and XLXG sub-units are only present at low abundance in the XyG of the *xlt2* mutant. However, the phenotypes of *xlt2* and wild type do not differ substantially (*Jensen et al., 2012*). The abundance of XyG in the walls of the *xxt1* and *xxt2* mutants, which have no visible morphological defects, is decreased by approximately 10% and 32%, respectively (*Cavalier et al., 2008*). These results suggest that structural variation of XyGs or difference in abundance of XyG does not necessarily alter a plants phenotype. Indeed, the *xxt1xxt2* double mutant, which has no detectable XyG in its cell wall, has only modest phenotypes including short root hairs and hypocotyls together with slightly reduced growth (*Cavalier et al., 2008*; *Park & Cosgrove, 2012*). By contrast, elimination of *AtMUR3* results in severe phenotypic changes. These plants have a cabbage-like phenotype with short petioles, curled rosette leaves, short etiolated hypocotyls and endomembrane aggregation phenotype (*Kong et al., 2015*; *Li et al., 2013*; *Tamura et al., 2005*; *Tedman-Jones et al., 2008*).

The *Arabidopsis MUR3* gene encodes a protein that is, evolutionarily related to animal exostosins, which are encoded by tumor-suppressor genes with roles in human bone growth (*Solomon, 1964*). *Arabidopsis* MUR3 transfers a galactosyl to the xylose adjacent to the unbranched glucose residue of XXXG and XLXG to form the XXLG and XLLG subunits (*Jensen et al., 2012*; *Madson et al., 2003*). The *mur3-1* and *mur3-2* mutants, which have a S470L or a A290V single point mutation, respectively, in the MUR3 protein are deficient in the α-L-fucosyl-(1→2)-β-D-galactosyl side chain (the XXFG and XLFG subunits) but have phenotypes similar to wild type plants (*Madson et al., 2003*; *Pena et al., 2004*). Further studies confirmed that *mur3-1* and *mur3-2* are leaky mutants with
discernible MUR3 activity as their XyG contain small amounts of XXFG and XLFG sub-units (*Kong et al., 2015*). Subsequently, two T-DNA insertion knock-out mutants *mur3-3* and *mur3-7* were identified and shown to have a dwarf cabbage-like phenotype (*Kong et al., 2015*; *Tamura et al., 2005*; *Tedman-Jones et al., 2008*). The XyG from *mur3-3* to *mur3-7* plants contains only XXXG and XLXG subunits (*Kong et al., 2015*). This indicated that the absence of F side chains results in the abnormal phenotype of *mur3-3* or *mur3-7*. However, the cabbage-like phenotype of *mur3-3* is rescued in the *xxt2mur3-3* and *xxt5mur3-3* double mutants, which also produce XyG comprised of only XXXG and XLXG subunits (*Kong et al., 2015*). The abundance of XLXG was almost double that of *mur3-3* XyG, which led us to suggest that a decrease of XyG galactosylation rather than the absence of the F side chain is responsible for the *mur3-3* cabbage-like phenotype (*Kong et al., 2015*). Indeed, overexpressing *AtXLT2* in *mur3-3* (35S*pro*:XLT2: *mur3-3*) resulted in a wild type phenotype and a XyG in which XLXG accounted for up to 85% of the subunits (*Kong et al., 2015*).

A previous study reported that rice *MUR3* (*OsMUR3*; Os03g05110) is functionally equivalent to *AtMUR3*. However, overexpression of *OsMUR3* in the *xlt2mur3-1* mutant only rescued the dwarf phenotype when XyG galactosylation was 81%. Three transgenic lines which produce XyG with 100% galactosylation did not (*Liu, Paulitz & Pauly, 2015*). Thus, XyG functions normally when the degree of XyG galactosylation exists within a certain range. Functional *MUR3* homologous genes have also been identified in eucalyptus, tomato, and sorghum (*Lopes et al., 2010*; *Schultink et al., 2013*; *Xu et al., 2018*). These homologs were reported to be functionally equivalent to *AtMUR3*, although differences in activity were discernible. For example, the *SlMUR3* (Sl09g064470) catalyzed galactosylation at the second xylose of the XXXG motif. The *AtMUR3* gene product does not (*Schultink et al., 2013*). In sorghum, overexpression of two homologous *AtMUR3* genes (*GT47_2* and *GT47_7*) only partially rescued the dwarf cabbage-like phenotype of *mur3-3* (*Xu et al., 2018*). Here, we report two homologous *AtMUR3* genes from *Brassica rapa* to *Brassica oleracea* that rescue the *mur3-3* phenotype.

## MATERIALS AND METHODS

### Gene identification and phylogenetic analysis

The protein sequence of *Arabidopsis* MUR3 was used as the query to perform BLAST analyses in the NCBI database with the Protein BLAST tool (https://blast.ncbi.nlm.nih.gov) to identify *MUR3* homologs in other plant species. Tobacco *MUR3* homologs were identified using http://solgenomics.net/organism/Solanum_lycopersicum/genome. Clustal W software was used to align the full-length protein sequences. MEGA 6 software (*Tamura et al., 2013*) was used to construct a phylogenetic tree with Neighbor Joining (NJ) parameter.

### Gene cloning and transformation

The coding sequences of candidate *AtMUR3* homologous genes were amplified with the primers listed in Supplemental File 1. The full length cDNAs were cloned into the *pCAMBia1300* overexpression vector (*Kong et al., 2009*) directly with Seamless Assembly

Cloning Kit (C5891; Clone Smarter, Houston, TX, USA). *Agrobacterium tumefaciens* strain GV3101 was used to introduce the constructs into *mur3-3* (*At2g20370*; Salk_141953) with the dip infiltration method (*Clough & Bent, 1998*). Transgenic plants were selected on one-half-strength Murashige and Skoog (1/2 MS) (*Murashige & Skoog, 1962*) plates containing 15 μg mL$^{-1}$ hygromycin. The T$_2$ lines were used for subsequent analysis.

## Plant growth conditions, phenotypic and genotyping

*Arabidopsis* plants were grown a 19 °C on soil with a 16-h-light and 8-h-dark cycle in environmental-controlled growth chamber (*Xu et al., 2017*). Seeds were surface sterilized and hypocotyl growth determined as described (*Xu et al., 2017*). Images of hypocotyls and adult plants were obtained with a Canon 5D Mark III digital camera. Hypocotyl length and plant heights were measured and then analyzed using Image J software (*Abramoff, Magalhaes & Ram, 2004*).

To identify the *MUR3* gene background in transgenic plants, DNA was extracted from *Arabidopsis* rosette leaves using the EasyPure Plant Genomic DNA Kit (EE111-01; TransGen, China). Total RNA was extracted from *Arabidopsis* rosette leaf, stem, hypocotyl and mature root using the EasyPure Plant RNA Kit (ER301-01; TransGen, China). TransScript®One-Step gDNA Removal and cDNA Synthesis SuperMix Kit (AT311-02; TransGen, China) was used to synthesize first-strand cDNA. This DNA was then used to determine the expression level of the homologous *MUR3* genes in transgenic plants by semi-quantitative RT-PCR. The *Arabidopsis ACTIN* gene was used as a reference. The primers used are listed in Supplemental File 1.

## Monosaccharide composition analysis

Rosette leaves of 4-week-old *Arabidopsis* plants (WT, *mur3-3*, independent complemented lines for *BrMUR3* and *BoMUR3* genes) were collected and used for cell wall monosaccharide compositions analysis as described (*Chai et al., 2015*). The alcohol insoluble residues (AIRs) were then prepared (*Xu et al., 2018*). In brief, leaf powder was sequentially extracted for 30 min each with aq. 70%, 80% and then 100% alcohol. The final residue was suspended for 2 h at 37 °C in acetone, filtered and air dried. The AIR was destarched using α-amylase and amyloglucosidase (Sigma–Aldrich, St. Louis, MO, USA).

The AIR was hydrolyzed for 2 h at 120 °C with 2 M trifluroacetic acid (TFA) to generate free monosaccharides (*Balaghi et al., 2011*; *Yu et al., 2014*). The hydrolysates were then reacted for 30 min at 70 °C with 1-phenyl-3-methyl-5-pyrazolone (PMP). The mixture was extracted three times with chloroform. The PMP-monosaccharides were analyzed with a Waters high performance liquid chromatography (HPLC) system, a 2,489 UV visible detector and a Thermo ODS-2 C18 column (4.6 × 250 mm) (*Xu et al., 2018*). Three biological replicates were employed per sample.

## MALDI-TOF MS analysis

To determine XyG subunit composition in WT, *mur3-3* and complementary transgenic plants, the destarched AIRs were treated with 4 M KOH (*Kong et al., 2015*).

The 4M KOH-soluble material was neutralized, dialyze and freezed dried. The 4M KOH-soluble material in 50 mM ammonium formate (pH 5) was then treated with XyG-specific endoglucanase (XEG, two units). Ethanol was added to 70% (v/v). The precipitate that formed was removed by centrifugation and the soluble fraction concentrated to dryness by rotary evaporation. The dried residue was dissolved in water and freeze-dried repeatedly to ensure removal of ammonium formate.

A Bruker Microflex spectrometer and workstation (Bruker, Billerica, MA, USA) were used for positive ion mode MALDI-TOF MS analysis (*Xu et al., 2018*). XyG oligosaccharides solutions (~1 mg/mL, five µL) were then mixed with 10 mM NaCl (five µL). An aliquot (one µL) of this mixture was then added to an equal volume of 0.1 M dihydroxybenzoic acid on the MALDI target plate. The mixture was concentrated to dryness with warm air. Mass spectra were collected by summing spectra from at least 200 laser shots.

# RESULTS

## *MUR3* homologous genes identification and phylogenetic analysis

Nineteen *MUR3* homologous genes were identified in three monocots and 16 dicots (Supplemental File 2). Their predicted amino sequences together with the sequence of the *Arabidopsis* MUR3 protein were used to construct a phylogenetic tree, which was divided into two clades. One of the clades contained only *Nicotiana tabacum MUR3*. The other sequences all clustered into a second clade (Fig. 1). The *MUR3* homologs identified from *Camelina sativa*, *Eutrema salsugineum*, *Raphanus sativus*, *B. rapa*, *B. oleracea* and *B. napus* were mostly closely to *Arabidopsis thaliana*.

## Functionally validation of two homologous genes of *MUR3*

To verify the function of selected *MUR3* homologous genes, the constructs containing the coding sequences of *R. sativus* (XM_018584632.1), *B. rapa* (XM_009103789.2), *B. oleracea* (XM_013742723.1), *Gossypium hirsutum* (XM_016854228.1) and *Nicotiana tabacum* (mRNA_42310_cds) were transformed into the *mur3-3* mutant. Complete rescue of the *mur3-3* mutant phenotype was obtained only in transgenic plants that overexpressed XM_009103789.2 (*BrMUR3*) and XM_013742723.1 (*BoMUR3*) (Fig. 2). The curled rosette leaves and short petioles phenotype of *mur3-3* are fully rescued by overexpression of *BrMUR3* and *BoMUR3* in complementary plants *BrMUR3_11* and *BoMUR3_22* (Figs. 2A–2D). Transgenic *mur3-3* mutants that overexpressed the *R. sativus*, *G. hirsutum* and *N. tabacum* homologs of *AtMUR3* retained the cabbage-like phenotype (Fig. S1).

The *mur3-3* mutants complemented with *BrMUR3* and *BoMUR3* had their size restored to normal compared to WT (Figs. 2A–2D). The height of adult *mur3-3* mutants is nearly 50% less than WT. The two complementary plant lines rescued this dwarf defect to WT level (Figs. 2E–2I). Additionally, the shorter hypocotyl phenotype of the *mur3-3* mutant were also be restored by overexpression of *BrMUR3* and *BoMUR3* (Figs. 2J–2P). These results indicate that *BrMUR3* and *BoMUR3* function equivalently to *AtMUR3*.

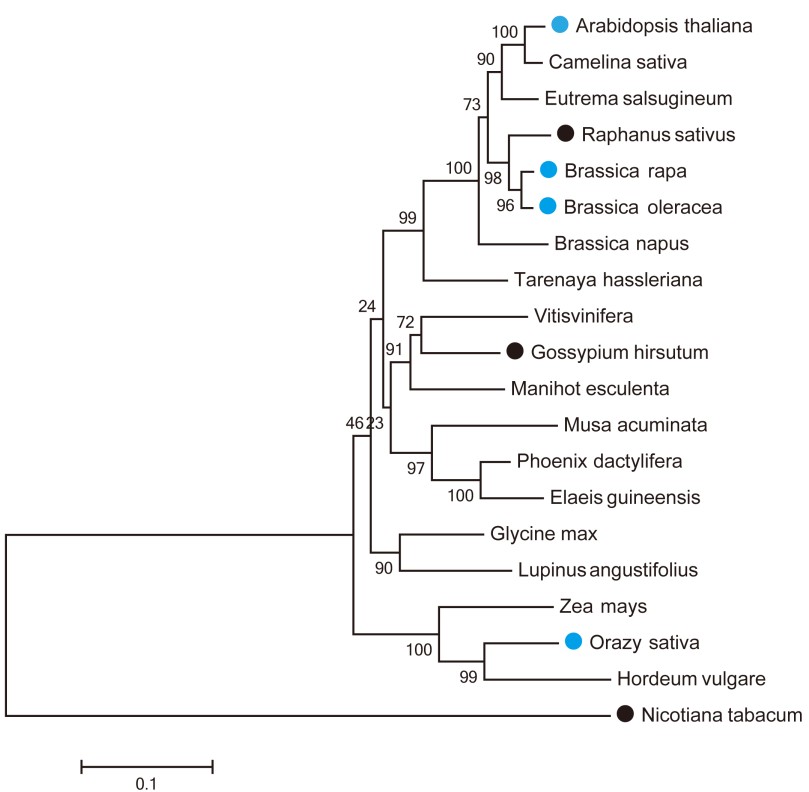

**Figure 1 Phylogenetic relationships of *AtMUR3* homologous genes.** The blue circle indicates the genes that rescued the dwarf phenotype of *mur3-3*; the black circle indicates the genes that did not rescue this phenotype.                

## Transcript identification of *BrMUR3* and *BoMUR3* in transgenic lines

Four independent $T_2$ homozygous transgenic lines that overexpress *BrMUR3* (*BrMUR3_10*, *BrMUR3_11*) or *BoMUR3* (*BoMUR3_21*, *BoMUR3_22*) in *mur3-3* were identified by semi-quantitative PCR. No transcript of *AtMUR3* was detected in six other complemented lines (*BrMUR3_7*, *BrMUR3_10*, *BrMUR3_11*, *BoMUR3_21*, *BoMUR3_22* and *BoMUR3_25)* indicating that they retained the *mur3-3* background (Figs. 3A and 3B). The expression of *BrMUR3* in *BrMUR3_10* and *BrMUR3_11* transgenic lines (Fig. 3C) and of *BoMUR3* in the *BoMUR3_21*, *BoMUR3_22* (Fig. 3D) was confirmed.

## Primary cell wall monosaccharide composition of *BrMUR3* and *BoMUR3* complementary plants

We next determined if the expression of *BrMUR3* and *BoMUR3* altered the monosaccharide compositions of the *mur3-3* cell wall. The fucose content is substantially reduced in the leaves of *mur3-3* compared with wild type (Table 1). The galactose content of *mur3-3* leaves is also decreased. No significant difference in relative abundances of arabinose, xylose, mannose, glucose, rhamnose and glucuronic acid were detected. The fucose and galactose contents were restored to wild type levels by ectopic overexpression of *BrMUR3* and *BoMUR3* in *mur3-3* (Table 1). These results strongly suggest that the BrMUR3 and BoMUR3 proteins have galactosyl transferase activity.

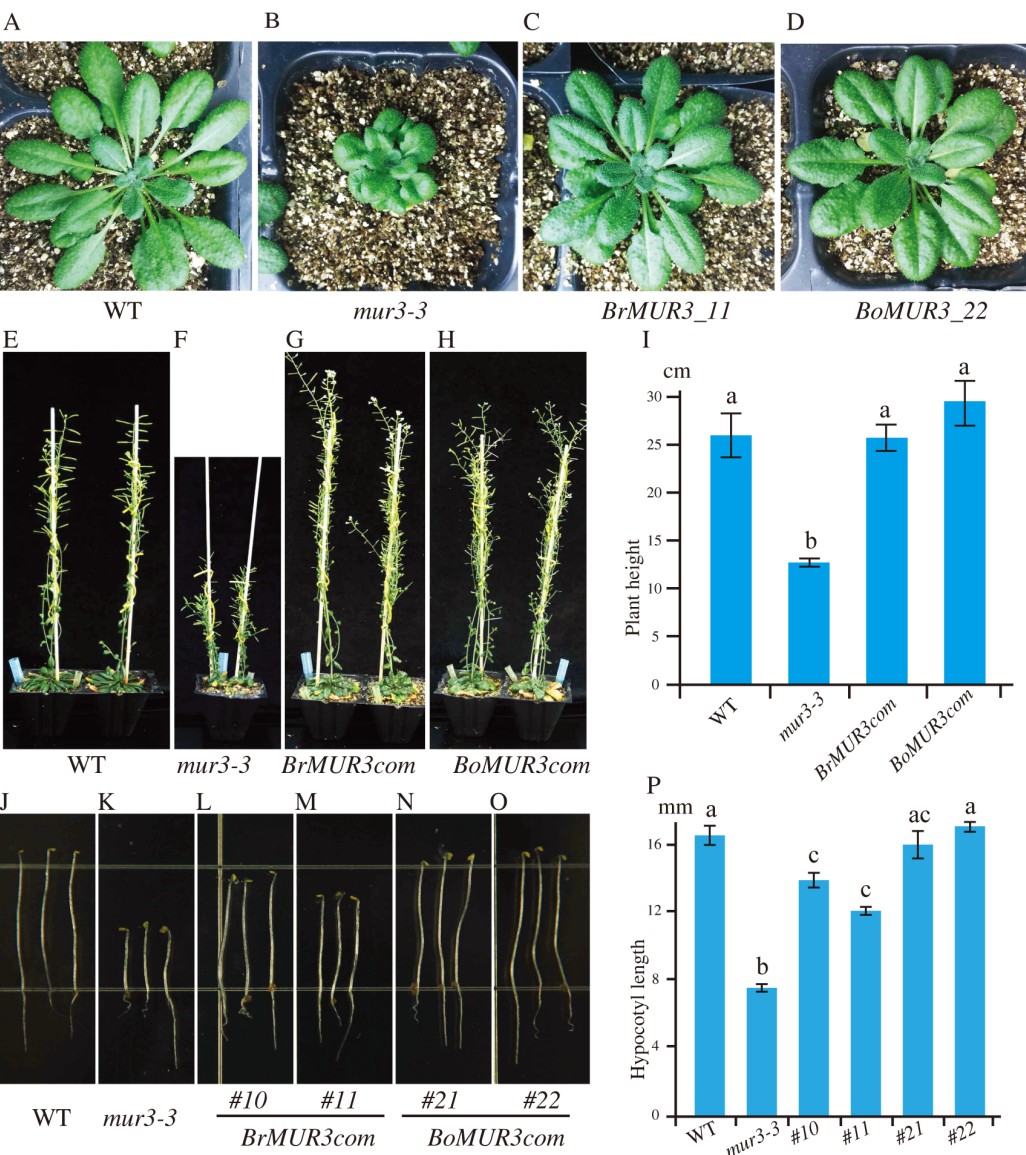

**Figure 2 Four-week-old aerial phenotypes of WT (A), *mur3-3* (B) and *mur3-3* plants overexpressing *BrMUR3* (C) and *BoMUR3* (D) and 8-week-old aerial phenotypes of WT (E), *mur3-3* (F) and *mur3-3* plants overexpressing *BrMUR3* (G) and *BoMUR3* (H).** The hypocotyl phenotype of WT (J), *mur3-3* (K) and the complemented lines overexpressing *BrMUR3* (L and M) and *BoMUR3* (N and O). The eight-week-old plants height (I) and hypocotyl length (P) of WT, *mur3-3* and *mur3-3* plants overexpressing *BrMUR3* and *BoMUR3* were statistics. For each genotype, at least 10 independent plants were randomly selected for height and hypocotyl length measurements. The column bars are the mean ± SD. The different letters indicate statistically significant differences from WT ($p < 0.05$).

## XyG structure analysis with MALDI-TOF MS

We determined if the expression of *BrMUR3* and *BoMUR3* affected the structure of the *mur3-3* XyG. The *mur3-3* XyG was comprised of XXXG (58%) and XLXG (42%). No XXLG, XXFG and XLFG subunits were detected in the *mur3-3* XyG (Fig. 4B). By contrast, ions corresponding to the XXLG/XLXG (*m/z* 1,247), XXFG (*m/z* 1,393),

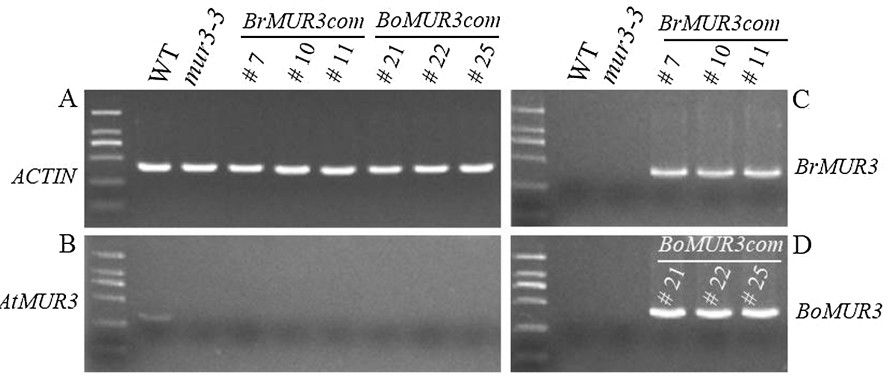

**Figure 3 Expression levels of *AtMUR3* (B), *BrMUR3* (C) and *BoMUR3* (D) in WT, *mur3-3* and complemented *mur3-3* plants.** Part (A) indicated the *ACTIN* gene expression level.

**Table 1 Glycosyl residue compositions of the cell walls from wild-type, *mur3-3* and complemented transgenic plants.**

| Samples | Man | Rha | Glc A | Gal A | GLC | Gal | Xyl | Ara | Fuc |
|---|---|---|---|---|---|---|---|---|---|
| mg per g AIR[1] | | | | | | | | | |
| WT | 12.8 ± 0.66a[2] | 5.0 ± 0.25a | 10.0 ± 0.38a | 76.2 ± 0.84a | 28.0 ± 0.91a | 33.8 ± 1.11a | 16.1 ± 0.83a | 21.6 ± 0.66a | 1.0 ± 0.05a |
| *mur3-3* | 12.4 ± 0.22a | 5.1 ± 0.24a | 10.3 ± 0.54a | 76.6 ± 3.42a | 27.3 ± 0.47a | 27.7 ± 1.58b | 16.6 ± 0.85a | 22.3 ± 0.53a | 0.6 ± 0.02b |
| *BrMUR3*com #10 | 12.7 ± 0.37a | 4.9 ± 0.13a | 9.9 ± 0.08a | 78.8 ± 1.49a | 27.9 ± 0.79a | 32.9 ± 1.20a | 16.9 ± 0.63a | 23.7 ± 0.60a | 1.2 ± 0.05a |
| *BrMUR3*com #11 | 12.5 ± 0.43a | 4.8 ± 0.13a | 10.2 ± 0.13a | 77.5 ± 1.11a | 28.8 ± 0.50a | 33.5 ± 0.65a | 16.8 ± 0.52a | 22.6 ± 0.64a | 1.1 ± 0.04a |
| *BoMUR3*com #21 | 12.3 ± 0.46a | 5.1 ± 0.22a | 9.9 ± 0.42a | 78.4 ± 1.57a | 29.2 ± 1.40a | 34.7 ± 1.04a | 16.7 ± 0.82a | 22.4 ± 0.14a | 1.2 ± 0.06a |
| *BoMUR3*com #22 | 12.9 ± 0.40a | 5.2 ± 0.27a | 10.1 ± 0.28a | 76.5 ± 4.19a | 30.4 ± 1.11a | 33.8 ± 1.49a | 16.5 ± 0.42a | 22.8 ± 0.23a | 1.3 ± 0.07a |

**Notes:**
[1] Values are mean ± SD ($n$ = 3).
[2] Means marked with the letter a are not significantly different from wild type. However, means marked with the letter b are significantly different from wild type (One-way ANOVA, Duncan's test, $p < 0.05$).

XLLG ($m/z$ 1,410) and XLFG ($m/z$ 1,555) subunits were detected in the XyG from *mur3-3* plants complemented with *BrMUR3* (Fig. 4C) and *BoMUR3* (Fig. 4D). The XXXG, XLXG/XXLG, XXFG, XLLG and XLFG subunits were present in the ratios of 1.0:1.2:1.7:0.5:1.4 and 1.0:0.6:0.8:0.4:0.5 in the *BrMUR3* and *BoMUR3* transformed lines, respectively. Thus, between 70% and 83% of the subunits contain galactose and between 40% and 53% contain fucose. In wild type XyG ~70% of the subunits are galactosylated and ~48% are fucosylated

# DISCUSSION

The MUR3 sub-clade of glycosyltransferase family GT-47 contains enzymes with diverse activities. These enzymes add a galactose, an arabinose, or a galacturonic acid residue to the first, second or third position of the XXXG motif (*Jensen et al., 2012*; *Madson et al., 2003*; *Peña et al., 2012*). A previous study reported that a new XyG oligosaccharide (minor peak between XXFG and XXXG) was produced by *mur3* plants expressing *OsMUR3* (*Liu, Paulitz & Pauly, 2015*). Thus, this enzyme family provides a unique opportunity to study factors that control the specificity of galactosyltransferase acceptors and donors.

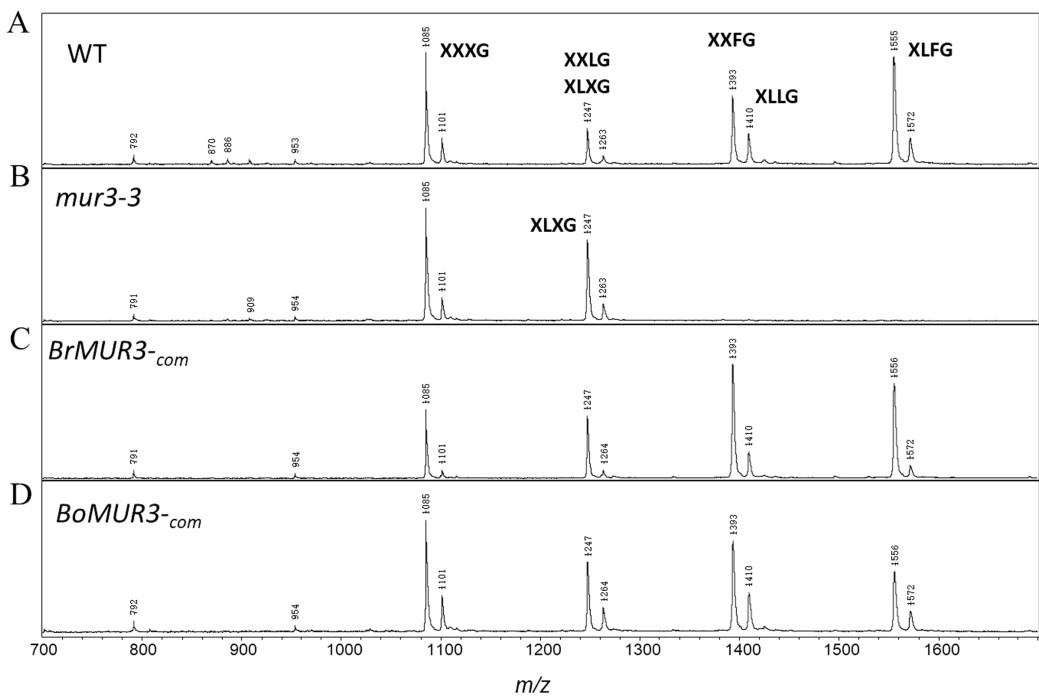

**Figure 4 MALDI-TOF mass spectra of the subunits generated from the XyG of 4-week-old WT (A), *mur3-3* (B), *BrMUR3*_com (C) and *BoMUR3*_com (D).** The Ions ((M + Na)$^+$) representing known XyG subunits are labeled using the one-letter code nomenclature (*Tuomivaara et al., 2015*).

Overexpression of the *B. rapa* and *B. oleracea* MUR3 homologs in *mur3-3* led to the formation of XyG containing XXFG and XLFG subunits (Figs. 4C and 4D). No new oligosaccharides structures were discernible. In *Arabidopsis* the F sidechain is formed by adding a fucosyl residue to the MUR3 Gal. This Gal is attached to the third Xyl. Fucosylation of the Gal attached to the second Xyl is rarely if ever observed. Thus, *BrMUR3* and *BoMUR3* and *AtMUR3* likely encode proteins with the same activity and substrate specificity. Furthermore, over expression of *BrMUR3* and *BoMUR3* rescue the cabbage-like phenotype of *mur3-3* (Figs. 2A–2D). We conclude that BrMUR3 and BoMUR3 are functionally equivalent to MUR3 and are orthologs of *AtMUR3*. Our results also show that XyG biosynthesis in brassicaceous vegetables involves *MUR3*-like genes.

The AtMUR3 protein was shown using in vitro activity assays to be a galactosyltransferase that specifically adds a Gal residue to the third position of the XXXG motif in XyG (*Madson et al., 2003*). Homologs of *AtMUR3* are present in many angiosperms, including nasturtium (*Jensen et al., 2012*) and eucalyptus (*Lopes et al., 2010*). The *Eucalyptus grandis* MUR3 protein has been reported to over galactosylate XyG (*Lopes et al., 2010*), suggesting it has enzymatic activities that differ from AtMUR3.

Mutant plants lacking the *MUR3* gene and the *xlt2mur3-1* double mutant are dwarfed and have curled rosette leaves. This phenotype is likely the consequence of a reduction of XyG galactosylation (*Jensen et al., 2012*; *Kong et al., 2015*). However, when these mutants are crossed with the *xxt1xxt2* double mutant that produces no XyG, the

homozygous offspring (*xxt1xxt2mur3-3*) produce no XyG and have a normal phenotype (*Kong et al., 2015*). These studies provide convincing evidence that normal growth is affected by altering the structure of XyG rather by the elimination of XyG. The altered XyG is thus a dysfunctional molecule (*Kong et al., 2015*).

Previous studies reported that overexpressing *OsMUR3*, tomato *MUR3* (*SlMUR3*), and *AtXLT2* in *xlt2mur3-1* or *mur3-3* rescued the "cabbage-like" phenotype of the mutants (*Kong et al., 2015*; *Liu, Paulitz & Pauly, 2015*; *Schultink et al., 2013*). However, the dwarf phenotype still existed in *OsMUR3* overexpression lines that showed over galactosylation of XyG (*Liu, Paulitz & Pauly, 2015*). Overexpression the *E. grandis* or tomato *MUR3* gene in *mur3* or *xlt2mur3-1* mutants also resulted in hyper galactosylation XyG (but not complete galactosylation). However, none of a dwarfed plant was reported in the transgenic plants (*Lopes et al., 2010*; *Schultink et al., 2013*). Thus, it appears that the phenotypic effect of altered XyG galactosylation varies with species. The factors that regulate the activity of XyG glycosyltransferases are not known. Presumably these activities must be coordinated since the structure of XyG is maintained within a plant.

*Brassica rapa* and *B. oleracea* and *Arabidopsis* are members of the family Brassicaceae, which typically produce XXXG type XyG (*Hoffman et al., 2005*). In this study, we have provided evidence that the *MUR3* genes from different *Brassica* encode enzymes with similar function with *AtMUR3*. The *AtMUR3* homolog identified in tomato (*SlMUR3*) also encodes a protein with similar XyG galactosyltransferase specificity (*Schultink et al., 2013*). However, it is notable that minor amounts of XLLG and XLFG were detected in addition to XXXG, XXFG, XXLG and XXFG, in the transgenic plants (*xlt2mur3-1*). Thus, *SlMUR3* may also catalyze the addition of Gal to the second position of XXXG.

Advances in our understanding of the catalytic mechanism and acceptor specificity of the MUR3 proteins will by facilitated by solving their crystal structures. The availability of functional recombinant versions of these GTs together with diverse acceptor molecules is also required to explore and define their substrate specifity. Such studies will form the basis for the production of GTs with new specificities and provide the opportunity to engineer polysaccharides with tailored functionalities.

## CONCLUSION

Understanding the functional significance and genetic basis of plant cell wall polysaccharide structural diversity remains a major challenge, Only a limited number of carbohydrate active enzymes involved in wall synthesis have been characterized in detail. Our study provides additional galactosyltransferases from brassicaceous species for investigating plant cell wall biosynthesis.

### Funding

This work was supported by the National Natural Science Foundation of China (31900276, 31670302, 31570670 and 31470291), the Doctor Foundation of Shandong

(ZR2019BC073), the First Class Grassland Science Discipline Program of Shandong Province, and the Taishan Scholar Program of Shandong (to G.Z.). The funders had no role in study design, data collection and analysis, decision to publish, or preparation of the manuscript.

## Grant Disclosures

The following grant information was disclosed by the authors:
National Natural Science Foundation of China: 31900276, 31670302, 31570670 and 31470291.
Doctor Foundation of Shandong: ZR2019BC073.
The First Class Grassland Science Discipline Program of Shandong Province.
Taishan Scholar Program of Shandong.

## Competing Interests

The authors declare that they have no competing interests.

## Author Contributions

- Meng Wang performed the experiments, prepared figures and/or tables, and approved the final draft.
- Zongchang Xu conceived and designed the experiments, performed the experiments, analyzed the data, prepared figures and/or tables, authored or reviewed drafts of the paper, and approved the final draft.
- Shuaiqiang Guo performed the experiments, prepared figures and/or tables, and approved the final draft.
- Gongke Zhou analyzed the data, prepared figures and/or tables, authored or reviewed drafts of the paper, and approved the final draft.
- Malcolm ONeill performed the experiments, authored or reviewed drafts of the paper, and approved the final draft.
- Yingzhen Kong conceived and designed the experiments, analyzed the data, authored or reviewed drafts of the paper, and approved the final draft.

## Data Availability

The raw data is available in the Supplemental Files and Supplemental Figures.

## Supplemental Information

Supplemental information for this article can be found online at http://dx.doi.org/10.7717/peerj.9095#supplemental-information.

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
