# Peer review of "Identification of two functional xyloglucan galactosyltransferase homologs BrMUR3 and BoMUR3 in brassicaceous vegetables"

_PeerJ, doi:10.7717/peerj.9095_

## Round 0.1 · original submission · Major Revisions

Please address carefully all comments and revise the paper substantially.

Reviewer 1 ·

Basic reporting

Structure of the manuscript is fine for the scientific content but english and syntax are way below scientific standard and requires improvement

Experimental design

Very good but may be improved by full characterization of the xyloglucan modification in mur3-3 mutant plant overexpressing the Brassica putative galactosyltrasferase.

Validity of the findings

Very good. Data are novels and interesting, worth publications. Discussion is of interest and data supported but impaired by poor syntax. Correction required.

Additional comments

The manuscript by Wang and co-authors described the identification and the functional characterization of two galactosyltransferases from brassicaceous plants acting on xyloglucan (XyG) hemicellulose.
The two genes identified and characterized are BrMUR3 and BoMUR3 from Brassica rapa and Brassica oleracea species, respectively. Galactosyltransferase activity of both genes was demonstrated using a classical functional reverse genetic study where the authors used the mur3-3 mutant from Arabidopsis thaliana which is normally deprived of galactosylated XyG structure. Successfully, using transgenic Arabidopsis thaliana plant over expressing BrMUR3 and BoMUR3, the authors could showed evidence that both genes were able to rescue the mur3-3 mutant back to WT phenotype, and using MALDI mass profiling they showed that XyG profiling was matching the WT.
1. As displayed, the experimental design is well suited to the study, has been performed rigorously with respect to the protocols employed, which makes the data set presented convincing and meaningful. Although I was satisfied and convinced by the demonstration by the authors of BrMUR3 and BoMUR3 genes encoding XyG galactosyltransferase activities, I was deceived by them not fully characterizing the galactosylated XyG in their transgenic lines. Indeed by combining beta-galactosidase treatment of the XyG from the transgenic lines with linkage analysis they would have clearly demonstrated the type of transfer catalyzed by their putative enzymes. One would also argue that an additional experiment demonstrating in-vitro transfer of UDP-Gal onto AIRs prepared from mur3-3 mutant would have nicely demonstrated the transferase activity of the MUR3 proteins from Brassica species which remains putative without such experiments.
Although the data set is convincing in its actual form, one could point out additional experiments remaining to be done before planning the crystallographic studies of the MUR3 proteins from Brassica.
2. Another point of pure form but of outmost importance is the quality of the scientific language that is too low for publication in its actual form. Sometimes English mistakes in word spelling or syntax were not too numerous, and one could follow the authors thinking but often writing of poor quality in the manuscript was a limitation for comprehensive understanding of the speculation by the authors. I will listed hereafter some suggestion of improvement of the introduction section but the discussion would need also to be re-written in its form ( no issue with the data interpretation or speculation that are supported by the data).
a. For example, it would more clear, line 89 if the authors remind the reader that xxt1 and xxt2 are altered for xylostransferase 1 or 2, as it was clearly said line 85 for xlt2 being altered for a galactosyltransferase.
b. Line 91 XyG variation does not necessarly lead to “spectacular” variation of plant phenotype would be more accurate.
c. Line 93 “shorter” instead of short.
d. Line 94, change “and a little smaller than WT” by “and the double mutant was a little smaller than WT” (more fragile as well in fact).
e. Line 102 modify ref (L, 1964) put the name of the author for this reference.
f. Line 114 this implies us that the defiency…check syntax of that sentence.
g. L117, which “further” KO that AtXXT2…change further or removed it.
h. L122 That’s means.. no “ ’s ” needed
i. L134 eucalyptus instead of eucalypts
j. L136 were functionally not were functional
k. L192 change ref M et al 1999 by the name of the author
l. L194 correct PH5 to pH5
m. L267 , why using upper case for GLYCOSYL TRANSFERASE?
n. L271 check syntax (possibly glycosyl….).
o. L276 because instead of due to ?
p. L294 check syntax of that sentence
q. L303 put double mutant where it is appropriate
r. L308 syntax to be checked
s. L310 to 314, I did not get the message clearly for that part
t. L316_317 are all belong brassicaceous? Check syntax
u. L323 change complementary for transgenic or other better word
v. L327 change need to needed
w. L335 functional to functionaly

Reviewer 2 ·

Basic reporting

The English language needs improvement so that an international audience can clearly understand the text. This includes the usage of past and present tense, use of plural or singular, etc. Examples are:
Line 1: Homologs instead of homologies
Line 34: …, which were renamed as BrMUR3….
Line 67: ..wiche are encoded by
Line 68: …are predicted to be….
Line 74….fucosyltransferase that can….
Line 79-82: check the grammar in these sentences
Line 82: …the basic oligosaccharide structure of XyG is (not was) XXXG….
Line 87: use the abbreviation XyG instead of Xyloglucan throughout the whole document
Line 90: reference format not korrekt - [8, 13]
Line 96, 116, 123 and more: cabbage-like or „cabbage-like“ – this should be consitent throughout the text.
Line 100: present tense
Line 101: …exostosins ecoded by…
Line 102, 179, 192: check reference format
Line 102-104: check „specificity“, galactosyl and xylosyl should be followed by residue or moeity
Line 95-126: the whole paragraph needs to be structured more clearly, for example in line 104ff it is stated that mur 3-1 und 3-2 are deficient of XXFG and XLFG, but in line 112 ff it is stated that they contain F side chains. In line 95 ff it is stated that dysfunction of AtMUR3 results in severe phenotypic changes, but in Line 107 it is stated that mur 3_1 and mur3-2 have similar phenotypes to wild type. Here, a more clear description of the mutants and their phenotypes is necessary.
Line 122: the WT only has 9% of XLXG according to Kong et al. 2015 – I am assuming the authors wanted to refer tot he mur mutants with the 38% proportion? However, here again it is necessary to specify which mutant they are refering to, as the mur3-3 has ~38% XLXG whereas mur3_7 has ~27% XLXG according to Kong et al. 2015.
Line 129: functionally – check if adverb, verb or adjective is correct here.
Line 138: check „…was not in AtMUR3 gene…“
Line 140: rescue, not rescued and report not reported
Line 163: I do not understand the word „acrial“ – maybe „aerial“? also applied to figure 2 legend
Line 164: Check sentence grammar and language
Line 198: …were used….
Line 263: „transfer“ not „transform“
Line 275: here is a cut and hard to follow from the reason of the study to suddenly that some plants are not familiar. This should be improved by stating the phylogenetic relationship first, then giving the reason for the choice of plant species.
Line302: overexpression of OsMUR3 still resulted in cabbage-like phenotype according to liu et al. 2015 – this sentence should be corrected accordingly.
The abstract and the introduction would benefit from a concise description of the reason for the study. This is currently only found in the discussion in Line 269-275, but should be stated earlier already.
Table 1 needs to be improved by 1) units for the cell wall sugars, b) information about statistics that were performed
Figure 4 legend needs to be improved regarding language. Also naming of the transgenic lines is not consitent: BrMUR3-com in the legend versus BrMUR3com in the figure. Also the number of the line is missing. „XyGOs“ should be „XyGs“.
Supplemental Figure 1 and 2: correct for spelling error (e.g. Oryza – Orazy)

Experimental design

The manuscript fits the scope of the journal. Based on already observed functional differences in MUR3 homologs it is of relevance to identify and characterize further MUR3 genes of different species to shed light on the biological variance in xyloglucan biosynthesis.
The methods could benefit of more detail and information for replication with respect to MALDI-TO MS analysis.
I would strongly recommend doing statistical analysis on Hypocotyl length of the MUR3 transgenic lines to see if brMUR3 can fully or only partially complement this phenotype.
In addition the results of the complementation with G.hisutum, N. tabacum and R. sativus should be presented as well even though they did not complement the mur3-3 mutant. However, they are relevant considering the reason of the study (investigating MUR3 homologs regarding specificity of glycosyltransferase activity) and worth showing in supplemental materials.

Validity of the findings

Analyzing the MUR3 clade of GT47 is of high relevance as the different mur3 Mutants in Arabidopsis already show differences in their phenotype and previous research has shown species-specific differences in the ability to complement the AtMUR3 mutant phenotypes. This indicates the biological variance and complex regulation of the xyloglucan biosynthesis pathway in plants. The authors identify 2 genes from Brassica species which can complement the mur3-3 phenotype they describe as cabbage-like.
In this respect the hypocotyl length analysis should be analyzed statistically (or repeated) to shed light if BrMUR3 can fully or only partially complement the short hypocotyl phenotype. From the pictures there seems to be only a partial complementation and this should be clarified and discussed.
The authors also discuss the phenotype complementation with regard to the degree of galactosylation in previous studies (Line 290-309). Therefore, it is important also to determine the proportion of galactosylation for the BrMUR3comp and BoMUR3comp transgenic lines. This is an analysis I would highly recommend in order to verify and justify this part of the discussion.
In line 256 the authors state that XXLG was also lost in mur3-3 mutant which is not in accordance with their figure 4 were the peaks for XXLG and XLXG are still present in mur3-3. This should be corrected.

Additional comments

The manuscript presents interesting new MUR3 genes from two brassica species that are shown to have similar functions as AtMUR3. However, analysis of galactosylation in the transgenic lines is missing which results in this part of the discussion standing alone, unconnected to the findings. I would highly recommend providing this data and include this in the discussion.
I would also recommend to include the negative results for the other three species in supplemental data, as those are still relevant regarding the reason of the study, to shed light on variance of the MUR3-clade of GT47.

---

## Round 0.2 · accepted · Accept

The authors correctly addressed the reviewer’s comments and suggestions and detailed these changes in the manuscript.

I believe that in the current form the article can be accepted for publication.